

# Testing the heat dissipation limitation hypothesis: basal metabolic rates of endotherms decrease with increasing upper and lower critical temperatures

Imran Khaliq[1,2,3] and Christian Hof[2,4]

[1] Department of Zoology, Ghazi University, Pakistan, Dera Ghazi Khan, Punjab, Pakistan
[2] Senckenberg Biodiversity and Climate Research Centre (BiK-F), Frankfurt, Germany
[3] Institute for Ecology, Evolution and Diversity, Department of Biological Sciences, Johann Wolfgang Goethe Universität Frankfurt am Main, Frankfurt, Germany
[4] Terrestrial Ecology Research Group, Department for Ecology and Ecosystem Management, School of Life Sciences Weihenstephan, Technical University of Munich, Freising, Germany

## ABSTRACT

Metabolic critical temperatures define the range of ambient temperatures where endotherms are able to minimize energy allocation to thermogenesis. Examining the relationship between metabolic critical temperatures and basal metabolic rates (BMR) provides a unique opportunity to gain a better understanding of how animals respond to varying ambient climatic conditions, especially in times of ongoing and projected future climate change. We make use of this opportunity by testing the heat dissipation limit (HDL) theory, which hypothesizes that the maximum amount of heat a species can dissipate constrains its energetics. Specifically, we test the theory's implicit prediction that BMR should be lower under higher metabolic critical temperatures. We analysed the relationship of BMR with upper and lower critical temperatures for a large dataset of 146 endotherm species using regression analyses, carefully accounting for phylogenetic relationships and body mass. We show that metabolic critical temperatures are negatively related with BMR in both birds and mammals. Our results confirm the predictions of the HDL theory, suggesting that metabolic critical temperatures and basal metabolic rates respond in concert to ambient climatic conditions. This implies that heat dissipation capacities of endotherms may be an important factor to take into account in assessments of species' vulnerability to climate change.

# INTRODUCTION

Understanding how organisms cope with their ambient climatic conditions is crucial for assessing whether and how species may be able to respond to ongoing and future climate change. Organisms' energy budgets play a key role in this context, and studies on how energy budgets vary in relation to environmental factors as well as to the species-specific characteristics will contribute to our knowledge about species' responses to varying and changing climatic conditions.

Corresponding author
Imran Khaliq,
imrankhaliq9@hotmail.com

Life history theory predicts that the energy available to organisms is limited and thus has to be utilized economically among different biological processes (*McNab, 2012*). An endotherm animal (i.e., a bird or a mammal) normally utilizes 30–50% of its daily energy expenditure just to maintain a stable body temperature (homeostasis) (*McNab, 2012*). In order to be able to invest sufficient energy into processes such as reproduction, dispersal or predator avoidance, animals should seek to live in environments with conditions allowing them to keep the energetic cost for homeostasis low (*Kobbe, Nowack & Dausmann, 2014*).

A non-reproductive endotherm at rest utilizes a minimal amount of energy within a range of temperatures known as thermal neutral zone (TNZ, *McNab, 2002*). Under the assumption outlined above, i.e., assuming an energetic trade-off between homeostasis and other biological processes, endotherms should live in areas with ambient temperatures falling within the TNZ (*Kobbe, Nowack & Dausmann, 2014*). However, studies have shown that for the majority of investigated endotherm species this expectation did not receive strong support (*Araújo et al., 2013*; *Khaliq et al., 2014*; *Khaliq et al., 2017*). This indicates that other physiological and ecological constraints shape the evolution, survival and distribution of endotherm animals (*Kearney & Porter, 2009*; *Fristoe et al., 2015*). In fact, in recent decades the identification of the factors that limit the energy availability of animals has become one of the most pursued topics in the field of eco-physiology (*Lovegrove, 2003*; *White et al., 2007*; *Wiersma et al., 2007*; *McNab, 2008*; *McNab, 2009*; *Wiersma, Nowak & Williams, 2012*). Researchers have proposed a plethora of factors that influence the energetics of animals (*White et al., 2007*; *White & Kearney, 2013*), in particular their basal metabolic rate (BMR), which is the minimal level of energy invested for homeostasis. On the one hand, several intrinsic factors, i.e., factors inherent to the animal's morphological or physiological characteristics have been shown to be associated with the variation in BMR (*Speakman et al., 2004*; *Wiersma, Nowak & Williams, 2012*; *White & Kearney, 2013*). For instance, species with small organ size tend to have lower BMR than species with larger organs (*Wiersma, Nowak & Williams, 2012*). As another example, the number of uncoupled mitochondria in individual mice have been shown to be positively associated with BMR (*Speakman et al., 2004*). On the other hand, BMR has been shown to be related to extrinsic factors, i.e., factors related to species' ambient environments. For example, BMR was negatively associated with ambient temperatures and rainfall variability (*Lovegrove, 2003*; *White et al., 2007*) and positively associated with the primary productivity (*Tieleman & Williams, 2000*; *Withers, Cooper & Larcombe, 2006*).

Traditionally, most studies have focused on extrinsic factors influencing the energy level in organisms, such as food sources (*McNab, 1988*; *McNab, 2009*) or environmental conditions (*Lovegrove, 2000*; *Lovegrove, 2003*; *White et al., 2007*). However, a few studies have also focused on intrinsic factors (*Johnson, Thomson & Speakman, 2001*; *Wu et al., 2009*; *Sadowska et al., 2016*). Regarding the latter, several hypotheses have gained attention. First, the central limitation hypothesis predicts that energy intake is constrained by the alimentary canal's capacity to process food, i.e., energy levels should be directly linked to the efficiency of the alimentary canal to convert food into energy (*Drent & Daan, 1980*; *Peterson, Nagy & Diamond, 1990*). In other words, available energy is constrained at the level of energy generation. However, this hypothesis has been questioned because it was

shown that under different climatic conditions animals ($n = 15$) can alter the rate of food processing beyond the upper limits set by the central limitation hypothesis (*Johnson & Speakman, 2001*; *Krol, Murphy & Speakman, 2007*). Second, the peripheral limitation hypothesis predicts that it is the internal organs' capacity to utilize energy that constrains the energy level in animals (*Hammond & Diamond, 1997*; *Bacigalupe & Bozinovic, 2002*). However, the observed possibility to alter energy utilization within the same individual animal (containing the same organs) under different climatic conditions has questioned this hypothesis as well (*Speakman et al., 2003*; *Renaudeau et al., 2012*).

Recently, as a third hypothesis on which intrinsic factors constrain the energy budget of endotherms, the heat dissipation limit (HDL) hypothesis has been put forward (*Speakman & Król, 2010*). It has received support from different lines of evidence; e.g., from observations that endotherms are able to alter energy levels under different climatic conditions (*Król & Speakman, 2003*) or that basal and field metabolic rates decrease with increasing ambient temperatures (*Speakman, 2000*; *White et al., 2007*; *Hudson, Isaac & Reuman, 2013*). These findings point towards the role of animals' heat dissipation capacity in limiting their energy levels. Further support for this comes from *McNab & Morrison (1963)* who showed that mammals inhabiting hotter environments are confronted with the endogenous heat load challenge due to their inefficiency to dissipate heat. When ambient temperatures rise, the difference between body and ambient temperatures decreases and it becomes increasingly difficult for endotherms to dissipate heat from the body. Consequently, it becomes a challenge to keep body temperatures constant. When temperatures rise beyond the upper limit of the TNZ (upper critical temperature, $T_{uc}$), the amount of heat generated at BMR exceeds the capacity of the body to dissipate heat passively, and other cooling mechanisms (e.g., evaporative cooling) must be employed to maintain body temperature and avoid hyperthermia.

To avoid an increased risk of hyperthermia under high and even increasing temperatures, endotherms may have two non-exclusive avenues of adaptation: changes in their thermal conductance (i.e., the ability to passively dissipate heat) or changes in their BMR as one crucial source of endogenous heat (*Naya et al., 2013*). All else being equal, to keep their body temperatures constant, species with a relatively high BMR must invoke active cooling mechanisms at a lower environmental temperature, i.e., have a lower $T_{uc}$, than species with a relatively low BMR. Similarly, species with a relatively high thermal conductance are able to dissipate more heat at a given environmental temperature than species with low thermal conductance. Thus, by increasing thermal conductance, decreasing BMR, or both, endotherms may increase their $T_{uc}$ (*McNab & Morrison, 1963*), hence being able to cope better with rising global temperatures (*Kobbe, Nowack & Dausmann, 2014*). Similarly, when environmental temperatures decrease, the difference between body temperatures and environmental temperatures increases and as a consequence the animal loses heat. When environmental temperatures reach the lower critical temperature ($T_{lc}$), the heat generated at the BMR level is not sufficient to keep body temperatures constant; therefore, the animal must invoke active heat generation (e.g., by increasing its food intake). Animals with relatively high BMR should therefore be able to extend their $T_{lc}$ towards low temperatures.

The HDL theory proposes that the overall energetics of endotherms are dependent on their abilities to dissipate heat from the body (*Speakman & Król, 2010*). There are a few studies that show lack of support for the HDL theory (*Petit, Vézina & Piersma, 2010*; *Wiersma, Nowak & Williams, 2012*); however, several studies in the recent past have empirically supported the HDL theory (*Brown, 1985*; *Speakman, 2000*; *Hinds & Rice-Warner, 2012*; *Hudson, Isaac & Reuman, 2013*; *Glazier, 2015*; *Sadowska et al., 2015*; *Woodroffe, Groom & McNutt, 2017*; *Zhang et al., 2018*). Still, a cross-species test of the HDL is lacking. Here, we test this prediction, using a data set of $T_{uc}$, $T_{lc}$, body mass, and BMR of 146 endotherms (88 mammals and 58 birds), in order to evaluate the influence of heat dissipation limits on species' metabolism. Following the above line of arguments, we expect, if heat dissipation is a limiting factor of endotherm energy levels (i.e., BMR), a negative relationship between critical temperatures and BMR.

## MATERIAL AND METHODS

### Data

Data for $T_{uc}$, $T_{lc}$, BMR and body mass for birds and mammals were compiled from published sources (see *Khaliq et al., 2014*; *Khaliq et al., 2015* for a detailed description). The compiled data were obtained from physiological experiments conducted to measure BMR under different temperature conditions. Data quality and suitability for macro-physiological analyses have been extensively debated recently (*Hof et al., 2017a*; *Hof et al., 2017b*; *McKechnie et al., 2017*; *Wolf et al., 2017*). Here, the focus on critical temperatures and BMR requires the use of data fulfilling the strictest quality criteria, i.e., a sufficient sample size (at least three individuals measured) and an experimental temperature range sufficient for determining $T_{uc}$ and $T_{lc}$ (at least three measurements of metabolic rate beyond TNZ) (see Table S1 in *Hof et al., 2017a*; *Hof et al., 2017b* for details). Thus, the dataset used here consists of 58 bird species (28 migrants and 30 residents) belonging to 28 families and 12 orders, and 88 mammal species belonging to 40 families and 16 order (see Table S1). To account for the evolutionary non-independence of data in comparative analysis, phylogenetic information for all species were compiled from published supertrees for birds (*Jetz et al., 2012*) and mammals (*Fritz, Bininda-Emonds & Purvis, 2009*; *Kuhn, Mooers & Thomas, 2011*).

### Analyses

To evaluate the influence of body mass on metabolic critical temperatures and to account for the joint evolutionary history of species, we used phylogenetic generalized least squares (PGLS) regression using the package *caper* (*Orme et al., 2012*) in R (*R Core Team, 2013*). This approach estimates a parameter λ (Pagel's lambda, *Pagel, 1999*), which indicates the amount of phylogenetic influence on the phenotype and applies a correction (*Martins, Hansen & Url, 1997*; *Freckleton, Harvey & Pagel, 2002*). We modeled $T_{uc}$ and $T_{lc}$ separately, using PGLS, as a function of log-transformed body mass while estimating λ by a maximum likelihood approach.

To test for the influence of heat dissipation on BMR, we modeled BMR as a function of body mass and then individually added $T_{uc}$ or $T_{lc}$ to the PGLS model, where $T_{uc}$ and

**Table 1** Phylogenetic generalized least squares models of BMR as a function of body mass and either upper or lower critical temperature ($T_{lc}$ or $T_{uc}$).

| | Birds ($n = 58$) | | | | Mammals ($n = 88$) | | | |
|---|---|---|---|---|---|---|---|---|
| | **B** | $\lambda$ | $R^2$ | **P** | **B** | $\lambda$ | $R^2$ | **P** |
| **Mass** | **0.70 ($\pm$0.03)** | – | – | **<0.001** | **0.68 ($\pm$0.01)** | – | – | **<0.001** |
| $T_{uc}$ | **−1.07 ($\pm$0.25)** | 0.66 | 0.89 | **<0.001** | **−0.57 ($\pm$0.23)** | 0.91 | 0.94 | **0.01** |
| **Mass** | **0.68 ($\pm$0.03)** | – | – | **<0.001** | **0.67 ($\pm$0.01)** | – | – | **<0.001** |
| $T_{lc}$ | **−0.31 ($\pm$0.13)** | 0.59 | 0.88 | **0.02** | **−0.32 ($\pm$0.12)** | 0.90 | 0.94 | **0.008** |

Notes.
BMR was first modeled using phylogenetic generalized least squares (*PGLS*), as a function of body mass while Pagel's $\lambda$ was estimated, and set it to its maximum likelihood value (see Methods). After controlling for phylogeny and body mass, we individually added either $T_{uc}$, $T_{lc}$ (log10-transformed) to the model. Bold values indicate associations where estimated parameters (B) are significantly different from 0. BMR and body mass were also log10-transformed. $R^2$ values refer to the full model.
$n$, sample size; $B$, estimated parameter $\pm$SE; $\lambda$, Pagel's Lambda, set to its maximum likelihood value.

$T_{lc}$ were used as proxies for species' heat dissipation abilities. To evaluate the robustness of the analysis across phylogenetic trees, we sampled 100 trees from the pseudo-posterior distribution of the used supertrees (*Kuhn, Mooers & Thomas, 2011*; *Jetz et al., 2012*) and ran the analyses for these 100 trees. For the final analysis we generated a maximum clade credibility (MCC) tree, using TreeAnnotator (included in BEAST v.1.7.5, *Drummond & Rambaut, 2007*).

# RESULTS

$T_{lc}$ and $T_{uc}$ co-varied with body mass (M), with the exception of bird $T_{uc}$ (Mammals: $T_{lc} = M^{-0.04}$, lambda = 1, $t = -3.002$, $p = 0.003$, $n = 88$; $T_{uc} = M^{-0.02}$, lambda = 0, $t = -4.6$, $p < 0.001$, $n = 58$; Birds: $T_{lc} = M^{-0.07}$, lambda = 0, $t = -3.13$, $p < 0.001$, $n = 58$; $T_{uc} = M^{-0.001}$, lambda = 0, $t = -0.16$, $p = 0.87$, $n = 58$). 8% of the variation in $T_{lc}$ and 19% of the variation in $T_{uc}$ of mammals could be explained by body mass. A higher amount of variation (13%) in $T_{lc}$ was explained by body mass in case of birds. As expected, body mass was positively related with BMR both in birds and mammals (Mammals: BMR = $M^{-0.69}$, lambda = 0.91, $t = 37.3$, $p < 0.001$, $n = 88$; Birds: BMR = $M^{-0.71}$, lambda = 0.66, $t = 19.2$, $p < 0.001$, $n = 58$). After accounting for the effect of body mass, $T_{lc}$ and $T_{uc}$ were negatively related with BMR both in mammals and birds (Table 1, Fig. 1).

# DISCUSSION

Our results show that endotherm species with higher mass-independent critical temperature values tend to have lower mass-independent BMR values, which is in accordance with our expectations based on the HDL theory. With higher $T_{uc}$ levels and lower mass-independent BMR under high ambient temperatures, endogenous heat load is reduced and hyperthermia can be avoided with lower efforts of evaporative cooling (*McNab & Morrison, 1963*; *Speakman & Król, 2010*). Similarly, under low ambient temperatures below $T_{lc}$ levels, heat is being lost more rapidly and to maintain body temperatures, heat generation above BMR levels is required. Thus there are two non-exclusive avenues to cope with lower temperatures: higher BMR levels or lower thermal conductance, e.g., via insulation. As a consequence, lower levels of $T_{lc}$ can be expected (*Scholander et al., 1950a*; *Scholander, 1955*).
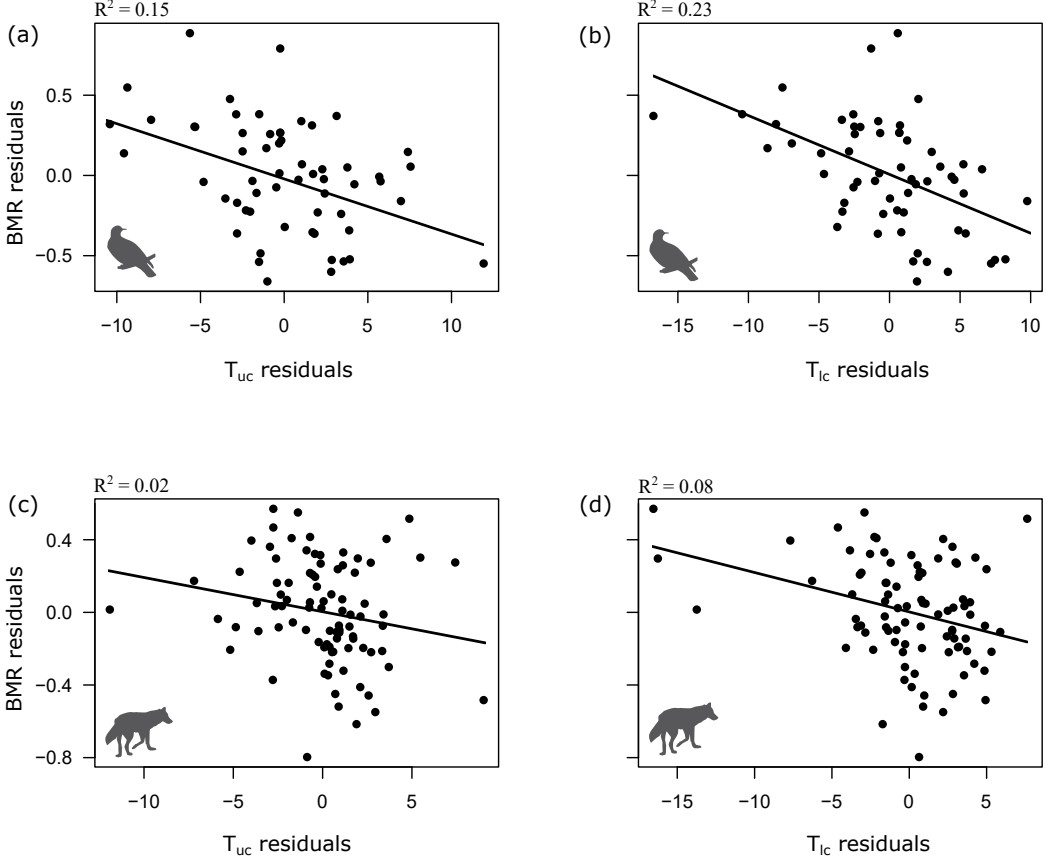

**Figure 1** **Relationship between BMR and upper and lower critical temperatures in birds (A, B) and mammals (C, D).** (A, B) birds, (C, D) mammals. BMR residuals are calculated as the BMR minus BMR as predicted by mass; and the $T_{lc}$ and $T_{uc}$ residuals are calculated as the $T_{lc}$ or $T_{uc}$ minus $T_{lc}$ or $T_{uc}$ as predicted by mass. All BMR, $T_{lc}$ and $T_{uc}$ values were log10-transformed before calculating the residuals. Note that these simple OLS analyses of the residuals only visualize the trends revealed by the full statistical model based on PGLS analyses (see Table 1 for details).

Our analyses accounts for the effect of body mass, which plays a central role in thermal physiology. Large endotherms tend to dissipate or lose metabolic heat slowly because of their small surface area relative to their body volume (*Scholander et al., 1950c*; *Riek & Geiser, 2013*). Thermal conductance scales with body mass with an exponent of around 0.57, while both BMR and body surface area scale with body mass with exponents around 0.66–0.74 (*Lovegrove, 2000*; *Lovegrove, 2003*; *White & Seymour, 2003*; *White & Seymour, 2004*; *McKechnie, Freckleton & Jetz, 2006*; *White et al., 2007*; *McNab, 2008*; *McNab, 2009*; *Jetz, Freckleton & McKechnie, 2008*; *Riek & Geiser, 2013*). This indicates that thermal conductance increases more slowly with body mass than BMR. Therefore, larger endotherms are better able to withstand colder temperatures than smaller endotherms (*Scholander et al., 1950b*). Similarly, small-bodied endotherms have a higher thermal conductance than larger species. Thus, small endotherms may be able to cope better with rising ambient temperatures than larger species (*Scholander et al., 1950b*). Overall the

interplay of body size, BMR and heat dissipation capacities allows endotherms to occupy a variety of environmental conditions (*Fristoe et al., 2015*).

The findings of the present study support the hypothesis that the capacity to dissipate heat is an important constraint for the energetics of endotherms (*Speakman & Król, 2010*). Recently, *Naya et al. (2013)* highlighted the importance of rodents' ability to dissipate heat in order to manage endogenous heat load and suggested that endotherm species (particularly non-tropical rodents) tend to increase their thermal conductance during the summer season (e.g., by seasonal changes in their insulation via fur). Such an increase in conductance should also be a means to increase their $T_{uc}$. Other studies also suggest that endotherm species do indeed raise their $T_{uc}$ during warmer months by lowering their basal metabolic rates (*Wilson, Brown & Downs, 2011*). Without such plastic changes in $T_{uc}$, endotherms may find it difficult to cope with the temporal variability in dry and hot conditions and as a result would frequently experience hyperthermia, which may have serious consequences for their survival (*McKechnie & Wolf, 2010*; *Speakman & Król, 2010*). Therefore, hot and dry conditions, as they are projected under future climate change scenarios for many regions, would probably favour species with lower mass-independent BMR and higher $T_{uc}$, while species with high BMR and, of course, those with lower $T_{uc}$ will face disadvantages.

All else being equal, it should be advantageous for endotherms to achieve extreme metabolic critical temperatures (i.e., very low $T_{lc}$ and very high $T_{uc}$) because wide thermal limits will allow endotherms to maintain BMR across a wide range of environmental temperatures, and thereby conserve energy (*Scholander et al., 1950b*; *Scholander et al., 1950c*). However, TNZs in endotherms are not infinitely wide; instead, $T_{lc}$ and $T_{uc}$ are positively correlated (*Riek & Geiser, 2013*; *Araújo et al., 2013*; *Khaliq et al., 2014*). This indicates that there are constrains for changing between low minimum conductance (i.e., low $T_{lc}$) and high maximum conductance (i.e., high $T_{uc}$) and thereby to alter the TNZ dynamically or adaptively to suit variable or changing climatic conditions. Specifically, in order to tolerate cold temperatures an endotherm requires a high energy level and low thermal conductance which is disadvantageous under higher ambient temperatures (*McNab, 2012*). This trade-off is supported by findings that the breadth of the TNZ of small endotherms varies seasonally which may be a mechanism to conserve energy (*Wilson, Brown & Downs, 2011*; *Kobbe, Nowack & Dausmann, 2014*).

Our findings on the relationship of metabolic critical temperatures with BMR may contribute to a better understanding of the relationships between species' energy budget and their environment, especially under changing climatic conditions. To off-set the negative effects of rising temperatures, endotherms may have at least two avenues for adaptation: (i) the alteration of heat dissipation capacities e.g., via altering body insulation (*Scholander et al., 1950a*), or (ii) achieving higher levels of metabolic critical temperatures. Endothermic species, particularly those in the tropics, might face the challenge of hyperthermia during periods of high ambient temperatures (see also *Khaliq et al., 2014*). Therefore, considering the heat dissipation capacities of endotherms may significantly improve assessments of species' vulnerability to climatic change.

## ACKNOWLEDGEMENTS

We thank Craig R. White, Diana Bowler, Meghan Balk, Markus Pfenninger, Stefan Ferger, and Humera Khaliq for commenting on previous versions of this paper. Comments by Amanda Bates and two anonymous reviewers greatly improved the manuscript.

### Funding

Imran Khaliq was supported by a doctoral scholarship by the Higher Education Commission of Pakistan. Furthermore, support was provided by the German Academic Exchange Service (DAAD) via its funding program "German-Pakistani Research Cooperations". Christian Hof was supported by the Bavarian Network of Climate Change "bayklif" of the Bavarian State Ministry for Science and the Arts. The funders had no role in study design, data collection and analysis, decision to publish, or preparation of the manuscript.

### Grant Disclosures

The following grant information was disclosed by the authors:
Higher Education Commission of Pakistan.
German Academic Exchange Service.
Bavarian Network of Climate Change.

### Competing Interests

The authors declare there are no competing interests. Christian Hof is an Academic Editor for PeerJ.

### Author Contributions

- Imran Khaliq conceived and designed the experiments, performed the experiments, analyzed the data, prepared figures and/or tables, authored or reviewed drafts of the paper, approved the final draft.
- Christian Hof analyzed the data, prepared figures and/or tables, authored or reviewed drafts of the paper, approved the final draft.

### Data Availability

The raw data and references are provided in the Supplemental Files.

### Supplemental Information

Supplemental information for this article can be found online at http://dx.doi.org/10.7717/peerj.5725#supplemental-information.

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
