# Peer review of "Testing the heat dissipation limitation hypothesis: basal metabolic rates of endotherms decrease with increasing upper and lower critical temperatures"

_PeerJ, doi:10.7717/peerj.5725_

## Round 0.1 · original submission · Major Revisions

Hello, Your reviewers found your article interesting, as did I. However we have several areas for improving the clarity of your manuscript, and I need further details on the data used (even if previously published) in order to make a decision. In addition to the comments provided by your reviewers, I have also attached an annotated pdf of your manuscript with a further set of comments aiming to improve the clarity and general readability of your manuscript. I will look forward to your revision - with best wishes, Amanda

Reviewer 1 ·

Basic reporting

Its good. But some minor errors and inappropriate citations - see comments to authors.

Experimental design

Statistical analysis of prior data so no design issues

Validity of the findings

It is good validated data well analysed.

Additional comments

The heat dissipation limit theory was published about 8 years ago and emerged from work in lactating mice which indicated that a major factor limiting their performance is the ability to get rid of body heat generated as a by-product of metabolic activity. It has resonances in earlier work from the 1960s. The theory makes several testable predictions one of which is that for animals that routinely experience high temperatures there should be selection to simultaneously increase the upper-critical temperature and decrease the basal metabolic rate. However because thermal conductance influences both upper and lower critical levels this will also lead to links between body mass and lower critical temperature, and between the two critical temperatures. This would then enable a greater metabolic scope at these high temperatures. In the current paper the authors test this idea by looking for an inverse correlation between estimates of BMR and the two critical temperatures in 146 endotherms. The results confirm the basic prediction of the HDL theory. The analysis is well performed and uses R to evaluate the phylogenetic signal in the data (which turns out to be unimportant). Have a few minor comments and suggestions.

Minor comments:

lines 41-42: This is not correct because of the possibility of heat substitution. That is energy derived as a by-product of exercise might be employed for thermoregulation purposes (see for example the review on this process by Humphries and Careau). The result is that costs of thermoregulation and exercise are not always additive.
Line 43: Reproduction does not enhance long-term survival. Indeed it may have the opposite impact. From an evolutionary perspective therefore animals should apply the level of EE that maximizes fitness.
Line 74: actually this work under cold conditions was presented in Johnston and Speakman 2001 (JEB) not Krol and Speakman 2003, which concerned exposure to hot conditions. Note that perhaps the strongest support comes from the shaving experiments by Krol et al (2007) also in JEB.
Lines 84: The paper by Speakman 1997 is not the best reference here as it contains rather limited data. Perhaps a better reference is the review by Speakman 2000 in Advances in ecological research 30: 177-197. There is a more direct independent evaluation of the HDL predictions with respect to DEE in Hudson et al 2013 (J animal ecology 82: 1009-1020)
Line 116: you should probably add the Hudson et al 2013 paper referred to above to this list of supporting evidence. Plus there is also lots of additional support, in studies of other lactating small mammals – see e.g. Sadowska et al 2016, timing of behavior (eg in wild dogs) and many other studies. If you scroll through the 200 or so papers that have cited the original presentation of the heat dissipation theory (Speakman and Krol 2010 in JAE) you will find many such examples of supportive literature. Indeed these outweigh the failures to support the idea many fold. Yet your presentation here seems to imply almost an equal balance of support and refutation.
Results: can you also show the relation between Tuc and Tlc (mass and phylogeny corrected) since this is also an important consideration and part of a heat trade-off as you point out later in the discussion. Since you have the data it would be nice to show it.
Line 175: ‘your expectations’ or in line with predictions of the HDL theory?
Line 176-178: there seems to be some mix up in the scales of description and analysis here. The analysis concerns interspecific trends in evolutionary time but this discussion seems to refer to individual animals expose to high temperature and their short term metabolic responses.
Line 190: there is of course a very large debate surrounding these numbers and I think for a more balanced approach you should cite a range with a few more pertinent citations.

Very minor points

Line 71: date missing from Drent and Daan 1980
Line 165: change tense to past
Line 168: change tense to past
Line 360: the initials for Speakman are incorrect

Reviewer 2 ·

Basic reporting

Please see my General comments for the Authors

Experimental design

Please see my General comments for the Authors

Validity of the findings

Please see my General comments for the Authors

Additional comments

The manuscript submitted by Imran Khaliq and Christian Hof is timely because it explores the correlation between basal metabolic rate (BMR) and thermoneutral zone (TNZ) defined by lower and upper critical temperatures (LCT and UCT, respectively) in mammals and birds, in the context of their thermal adaptations to different climate conditions and the factors limiting their energy expenditure and performance. The mechanisms responsible for thermal adaptations are central to understand how endotherms respond to climate change.

Despite being potentially interesting, there many aspects of the paper that are confusing and somehow under-explored or not properly explained. The Authors aim to test the heat dissipation limit (HDL) hypothesis, by exploring its prediction that ‘BMR should be lower under higher metabolic critical temperatures’ (lines 26-27). This is a simplification as the exact position of LCT and UCT depends on the interplay between heat production (represented by BMR) and heat loss (represented by thermal conductance). The Authors completely ignore thermal conductance and focus on the correlation between BMR and LCT/UCT, thus testing the prediction that has never been generated by the HDL hypothesis. The Authors claim that both central and peripheral limitation hypotheses are inferior to the HDL hypothesis because ‘under cold conditions animals can elevate energy levels’ (lines 73 and 78). Please elaborate which aspects of energy expenditure are elevated, otherwise it is rather difficult to understand the difference between central vs peripheral limits and the reasoning behind the HDL idea, which is supposed to be tested in the ms. There is no such thing as ‘energy creation’ (line 72) – please correct accordingly (‘Energy cannot be created or destroyed, it can only be changed from one form to another’, see the work of Albert Einstein).

It is great to have the raw data (Supplementary Table 1), but these data have no units, so are of limited use. Both BMR and thermoregulatory curve are prone to seasonal changes and shifts (e.g., Stawski et al. 2017 on bank voles, S. Eryn McFarlane et al. 2018 on flycatchers, Levesque et al. 2018 on Tupaia). Do the Authors filter their data accordingly and use only one-season data to reduce the variability observed in Fig. 1?

I was rather surprised to read the assumption made by the Authors that ‘species with lower metabolic critical temperatures should be able to dissipate heat more efficiently than species with higher critical temperatures’ (lines 151-153). First of all, with low UCT, there will be more cases that Ta > UCT, which makes the animal prone to overheating. Secondly, the animals with low critical temperatures tend to have higher BMR (Fig. 1), so they are expected to have more problems with overheating, because the costs of maintenance are higher (?). What happens to the significance in Fig. 1d, after removing 3 data points from the top left corner? Please check Table 1 for typing errors.

Taking into account all these points, I do not think that the current version of the manuscript is suitable for publication in PeerJ.

---

## Round 0.2 · accepted · Accept

I appreciate your efforts to revise your manuscript and am happy to accept; even so I do disagree with focusing on warming only in the conclusions (which is simply the research trend of the time which I can see is shifting to extreme events/seasonal variability) - however, I have decided to accept this version which narrowly pitches your work as this is ultimately your decision.

# Reviewer 1 ·

Basic reporting

I am happy with the response of the authors to my review and the changes they have made

Experimental design

no comments to add

Validity of the findings

no comments to add

Additional comments

I think this is now acceptable